# Reduction in the Dietary VA Status Prevents Type 2 Diabetes and Obesity in Zucker Diabetic Fatty Rats

**DOI:** 10.3390/biom12040528

**Published:** 2022-03-31

**Authors:** Tiannan Wang, Xia Tang, Xinge Hu, Jing Wang, Guoxun Chen

**Affiliations:** 1Department of Nutrition, University of Tennessee at Knoxville, Knoxville, TN 37996, USA; twang14@vols.utk.edu (T.W.); xhu25@vols.utk.edu (X.H.); 2College of Food Science and Technology, Hebei Agriculture University, Baoding 071001, China; bdzzp@163.com; 3College of Pharmacy, South-Central University for Nationalities, Wuhan 430074, China; 18772842249@163.com

**Keywords:** vitamin A, type 2 diabetes, Zucker diabetic fatty rats, oral glucose tolerance test, blood glucose, hepatic protein expression

## Abstract

We hypothesized that the vitamin A (VA) status regulates type 2 diabetes (T2D) development in Zucker diabetic fatty (ZDF) rats. Zucker Lean and ZDF rats at weaning were fed a VA deficient with basal fat (VAD-BF, no VA and 22.1% fat energy), VA marginal with BF (VAM-BF, 0.35 mg retinyl palmitate (RP)/kg), VA sufficient with BF (VAS-BF, 4.0 mg RP/kg), VAD with high fat (VAD-HF, 60% fat energy), VAM-HF or VAS-HF diet for 8 weeks, including an oral glucose tolerance test (OGTT) at week 7.5. The hepatic mRNA and proteins levels were determined using real-time PCR and Western blot, respectively. The VAD-BF/HF and VAM-BF/HF diets prevented peripheral hyperglycemia and attenuated obesity in ZDF rats, which occurred in the presence of the VAS-BF/HF diets. This lowered VA status reduced venous blood hyperglycemia, hyperinsulinemia and hyperlipidemia, and improved OGTT and homeostasis model assessment for insulin resistance results in ZDF rats. The expression levels of key hepatic genes for glucose and fat metabolism were regulated by VA status and dietary fat contents. An interaction between VA and HF condition was also observed. We conclude that the reduction in the dietary VA status in both BF and HF conditions prevents T2D and obesity in ZDF rats.

## 1. Introduction

Diabetes has become a concern of public health; as of 2019, there were 463 million adults with diabetes globally, based on the estimation of the International Diabetes Federation [1]. This led to a USD 760 billion health expenditure associated with diabetes worldwide in 2019 [2]. The annual health expenditures on diabetes in the United States and China are about USD 294.6 and 109.0 billion, respectively [2]. As patients with type 2 diabetes (T2D) account for 90% of the population, it is urgent to find ways to prevent and treat T2D. As both genetic and environmental factors contribute to T2D development [3,4], nutritional and lifestyle interventions are considered the first line of treatment for T2D before medications [5,6]. The daily intake of foods containing macronutrients and micronutrients change nutrients and hormone levels in the body. The disruption of this homeostasis may lead to the development of diseases. Therefore, a clear understanding of the roles of nutrients in T2D development is critical for its prevention and management.

Vitamin A (VA, retinol) as a micronutrient plays a very important role in a variety of physiological functions, including glucose and fat metabolism [7,8]. Body weight, glucose and lipid metabolism are regulated by VA status. For example, VA was discovered due to its role to support body weight gain in rats [9]. When rats were fed a VA-deficient (VAD) diet for 8 weeks after weaning (3 weeks of age), their body weight gain stopped [9]. Using the same feeding protocol, Zucker lean (ZL) and Zucker fatty (ZF) rats were fed a VA-sufficient (VAS) or VAD diet for 8 weeks in our lab [10,11,12,13]. We observed the depletion of body fat in ZL rats, and the prevention of obesity development in ZF rats fed with the VAD diet. The hepatic glycogen content was depleted in rats fed with the VAD diet [14], whereas in rats fed with a diet containing excessive retinyl ester, the hepatic glycogen level increased significantly [15]. When the energy intake of the VAD rats is adjusted via a pair-feeding experiment, the ZL and ZF rats fed the VAS diet with the same number of calories still have a higher body weight than those fed the VAD diet [10]. We have also shown that the VA status regulates the respiratory exchange ratio in ZL rats [13], showing the role of VA in the regulation of metabolism.

The development of T2D is associated with profound changes of glucose and lipid metabolism [16], which can be partially explained by the changes in insulin-regulated gene expression [17]. To find molecules that can modulate insulin-regulated gene expression in the liver, we extracted the lipophilic molecules from rat liver tissues and analyzed their impacts on the insulin-regulated expressions of the cytosolic form of the phosphoenolpyruvate carboxyl kinase gene (*Pck1*) and glucokinase gene (*Gck*) in primary hepatocytes, the key players in the hepatic gluconeogenesis and glycolysis, respectively. We reported that retinoids in the lipophilic extract modulated the insulin-regulated expression levels of *Pck1* [18] and *Gck* [19] genes. Later, we determined the retinoic acid responsive elements (RAREs) on the *Pck1* and *Gck* promoters in primary hepatocytes [20,21]. VA signaling is largely mediated by its metabolite, retinoic acid (RA). Additionally, we have shown the synergy between RA and insulin to induce the expression of the sterol regulatory element-binding protein 1c (*Srebp-1c)* gene [22], and identified the RAREs on its promoter [23]. All these demonstrate the role of VA signaling in the regulation of the expression levels of genes for hepatic glucose and lipid metabolism.

It is clear by now that VA participates in the regulation of glucose and lipid metabolism [24,25], suggesting that VA might have a role in the development of diabetes [8]. In human studies, T2D subjects have been associated with less VA among South Asians in the U.S. [26] or no difference of VA intake or plasma VA levels in those in the Alberta region of Canada [27] compared with the controls. The association of blood retinol level and T2D is still not conclusive [28]. It appears that dietary VA intake and plasma VA levels in T2D subjects vary in different populations [8]. In animal studies, RA treatment has been shown to reduce blood glucose levels in Zucker diabetic fatty (ZDF) rats (10mg/kg RA daily) for 2 weeks [29], and in female *ob/ob* mice (100 µg/mouse RA daily) for 16 days [30]. ZDF rats carrying a mutation of the leptin receptor have uncontrolled appetite, which leads to overnutrition and development of obesity and T2D [31,32]. It appears that RA production will benefit the control of blood glucose in rodents.

Therefore, we hypothesize that the VA status regulates the development of T2D and obesity in ZDF rats. To investigate the effects of VA status on T2D development, ZL and ZDF rats at weaning were fed a VAD, a VA marginal (VAM) [33] or a VAS diet containing a basal fat (BF) or high fat (HF) content for 8 weeks after weaning (3 weeks). We report here that the reduction in dietary VA content prevented the development of obesity and T2D in ZDF rats. These are associated with the improvement in insulin sensitivity and alterations of the expression levels of the hepatic genes involved in glucose and lipid metabolism.

## 2. Materials and Methods

### 2.1. Reagents

BD Vacutainer tubes, K3 EDTA (K3E) coated vacuum blood collection tubes, were obtained from BD (Franklin Lakes, NJ, USA). The insulin (FIN, Cat. # SRI-13K) and glucagon (Cat. # GL-32K) RIA kits were purchased from the Millipore Corporation (Billerica, MA, USA). Agarose was obtained from VWR International (Radnor, PA, USA). Rat leptin ELISA kit was purchased from RayBiotech (Norcross, GA, USA). DNA-free and cDNA synthesis kits, and 2 × SYBR Green real-time PCR Master Mix were purchased from Life Technologies (Grand Island, NY, USA). McKesson TURE METRIX Blood Glucose Meter and McKesson TURE METRIX test strips (McKesson Medical-Surgical Inc., Richmond, VA, USA) were used to measure the whole blood glucose level. The human insulin used in the insulin tolerance test (ITT) was obtained from Novolin^®^ R (100 units/mL, U-100, NDC 0169183302). The colorimetric kits for the plasma levels of glucose (catalog #:1070-500), cholesterol (catalog #: 1010-430), and triacylglycerol (TAG, catalog #: 2200-430) were obtained from STANBIO currently known as EKF Diagnostics (Borne, TX, USA). The bicinchoninic acid (BCA) protein assay kit was purchased from Pierce Biotechnology, Inc (Rockford, IL, USA) to test the protein concentration. Immobilon-PSQ PVDF membrane was purchased from EMD Millipore Corp (Billerica, MA, USA). Non-fat dry milk of Food Club was purchased from a local grocery store and used to block membranes in Western blot experiments. Protease-free bovine serum albumin and ECL Western Blotting substrate were obtained from Fisher Scientific (Waltham, MA, USA).

Antibodies to total fatty acid synthase (FAS, Cat #3180 s), ATP citrate lyase (ACL, Cat #4332 s), and β-actin (Cat #4970 s) and goat anti-rabbit IgG conjugated to horseradish peroxidase (#7074) secondary antibody were obtained from Cell Signaling Technology (Danvers, MA). Antibodies to the cytosolic form of phosphoenolpyruvate carboxykinase (PEPCK-C, #10004943) and glucokinase (GCK, sc-7908) were purchase from Cayman Chemical (Ann Arbor, MI) and Santa Cruz Biotechnology (Dallas, TX, USA), respectively. Insulin receptor β subunit (IRβ #610109) was obtained from BD Transduction Laboratories. ECL Western blotting substrate was purchased from Pierce Biotechnology, Inc (Rockford, IL, USA), which was used to visualize the recognized proteins.

### 2.2. Animals

To establish a breeding colony of ZDF rats, male and female heterozygous Zucker rats aged 35 days (Strain code, 380 *fa/+*) were obtained from Charles River (Wilmington, MA, USA). They were paired as one male to one or two female(s) per cage and fed a Teklad rodent chow (#8604, Envigo, Madison, WI, USA). Breeding pairs were housed in the animal facility in the Department of Nutrition at University of Tennessee, Knoxville on a 12 h light–dark cycle under constant temperature and humidity. Male ZL (*fa*/+ or +/+) and ZDF (*fa/fa*) rats at three weeks of age were weaned and used in the dietary study as indicated below. All procedures were approved by the Institutional Animal Care and Use Committee at the University of Tennessee at Knoxville (Protocols #1256, and #2647).

### 2.3. Rat Genotyping

Genotypes of ZL (*fa*/+ or +/+) and ZDF (*fa*/*fa*) rats were determined using a PCR-based procedure as developed in [34]. To obtain genomic DNA for analysis, ear tissue samples of rats aged at least 14 days were collected through ear punching. Seventy-five µL of alkaline lysis buffer (25 mM NaOH/0.2 mM EDTA) was added to each tissue sample, and then the mixture was incubated at 95 °C for 30 min. After that, 75 µL of 40 mM Tris HCl buffer was added. The mixture was vortexed and spun at 20,000× *g* for 5 min to obtain the supernatant tissue lysate containing the genomic DNA, which was used as the PCR template. Forward 5′-CGTATGGAAGTCACAGA-3′ and reverse 5′GAATTCTCTAAATATTTCAGC-3′ primers were designed to amplify a 101 bp of *fa* allele in the genomic DNA fragment flanking the rat *Lepr* mutation. The reverse primer was designed to anneal close to the mutation site and introduce a single base substitution (C to G) in the second base at the 3 ‘end of the primer to create a Pvu II restriction enzyme site only in the wild type, but not the mutant allele.

Each PCR reaction with a 25 µL volume contained the following, 2.5 µL tissue lysate template, 2.5 µL 10 × PCR buffer, 2.5 µL 25 mM MgCl_2_, 1 µL 100 mM dNTP, 0.5 µL 5 U/µL Taq DNA polymerase, 1 µL each of 10 µM forward and reverse primers, and 14 µL H_2_O. PCR amplification started at 95 °C for 5 min, which was followed by 40 cycles of 95 °C for 30 s, 60 °C for 30 s, and 72 °C for 30 s. Twenty µL of the PCR product were digested with 10 units of Pvu II (New England Biolabs) in a 28 µL reaction volume for up to 1 h at 37 °C. After digestion, 20 µL of the digested product were mixed with 5 µL loading buffer (60% sucrose) and were loaded onto a 4% agarose gel and electrophoresed in 0.5 × Tris-borate-EDTA solution in the presence of 10 mg/mL ethidium bromide. The wild-type genotype (+/+) showed the presence of an 80 bp band, whereas the mutant genotype (*fa*/*fa*) showed the presence of a 101 bp band. The heterozygote (*fa*/+) genotype showed the presence of both bands.

### 2.4. Diets and Experimental Groups

Male ZL (*fa*/+ or +/+) and ZDF (*fa/fa*) rats at weaning (3 weeks of age) were divided into 6 dietary groups with different VA statuses and dietary fat contents as shown in Table 1. The custom-made and synthetic BF and HF diets with different VA statuses were formulated and purchased from Envigo (#TD. 190036-190041). The BF diets contained around 19.7%, 57.9% and 22.5% of energy (3.8 kcal/g) from protein, carbohydrate, and fat, respectively. The HF diets contained around 18.5%, 21.4% and 60% of energy (5.1 kcal/g) from protein, carbohydrate, and fat, respectively. These diets were VAD with BF (VAD-BF, 0 mg retinyl palmitate/kg diet), VAM with BF (VAM-BF, 0.35 mg retinyl palmitate/kg diet) [33], VAS with BF (VAS-BF, 4.0 mg retinyl palmitate/kg diet), VAD with HF (VAD-HF, 60% energy from fat), VAM with HF (VAM-HF) and VAS with HF (VAS-HF) diets. ZL and ZDF rats after weaning (3 weeks of age) were fed one of these six diets for 8 weeks. During the feeding period, rats were housed at 1 to 3 per cage depending on the number and genotypes of rats available at weaning. The body mass (BM), total food intake, and peripheral blood glucose (PBG) were measured weekly during the same period.

### 2.5. ITT and Oral Glucose Tolerance Test (OGTT)

For the ITT and OGTT, rats were fasted overnight. The ITT was performed after the animals were fed their respective diets for 6.5 weeks (44 days after weaning). Short-acting insulin (Novolin^®^ R insulin human injection, U-100) was administrated to ZDF (0.25 U/kg BM) and ZL (0.125 U/kg BM) rats intravenously as indicated in [35]. After the insulin administration, tail tip whole blood samples were collected via percutaneous punches at 0 (the baseline data), 5, 10, 15, 20, 30, 40, and 60 min. The OGTT was performed after the animals were fed their respective diets for 7.5 weeks (51 days after weaning). After the overnight fasting, a glucose solution was administered to ZDF and ZL rats at a dose of 2 g/kg BM via gavage as described in [36]. After that, tail blood samples were collected via percutaneous punches at 0 (the baseline data), 10, 20, 30, 60, and 120 min. The whole blood glucose level was determined using McKesson glucometer and strips.

### 2.6. Plasma and Tissue Sample Collection and Measurements

At the end of the 8-week feeding protocol, rats were fasted for 6 h and euthanized via CO_2_ inhalation for the collection of plasma and tissue samples. Accumulated venous blood was collected using the K3 EDTA-coated BD vacuum blood collection tubes and placed on ice before further processing. Whole blood samples were centrifuged at 2000× *g* for 30 min at 4 °C. The supernatant (plasma) was transferred to fresh microcentrifuge tubes for storage. The liver was collected after the accumulated venous blood was drained. The two pads of epididymal white adipose tissue (WAT) were removed carefully. The liver and WAT tissues of each rat were weighed, and immediately frozen in liquid nitrogen. The plasma and tissue samples were stored at −80 °C until being used for further analysis.

The plasma concentrations of blood glucose, triacylglycerol and total cholesterol levels were determined using kits from STANBIO Laboratory following the manufacturer’s manual. The plasma insulin and glucagon levels were measured using the RIA kits according to the manufacturer’s manuals. The plasma leptin level was measured using the ELISA kit following the manufacturer’s manual.

The homeostasis model assessment for insulin resistance (HOMA-IR) was calculated with the formula: HOMA-IR (nIU/mL × mM) = fasting blood glucose (mmol/L) × fasting insulin (μU/mL)/22.5. In this case, 22.5 is the constant of glucose turnover when glucose is expressed in insulin sensitivity index units.

### 2.7. RNA Extraction and Quantitative Real-Time PCR

Total RNA was extracted from about 100 mg frozen rat liver sample using 1 mL of TRIzol reagent (#98903, Life Technologies, Carlsbad, CA, USA) according to the manufacturer’s protocol. The quality of the RNA was determined using the ratio of absorbance at 260 nm and 280 nm. Only samples with a ratio higher than 1.8 were used. After the removal of the contaminated DNA, 2 μg of DNA-free RNA was used for cDNA synthesis. For the quantitative real-time PCR, oligo nucleotide primer sequences for *Gck*, sterol regulatory element-binding protein 1 (*Srebp-1c*)*,* sterol regulatory element-binding protein 2 (*Srebp2*), *Pck1*, fatty acid synthase (*Fas*)*,* and ATP citrate lyase (*Acl*) genes were used and are shown in the Appendix A. Each SYBR green based real-time PCR reaction contained, in a final volume of 14 μL, cDNA from 14 ng of reverse transcribed total RNA, 2.33 pmol primers, and 7 μL of 2 × SYBR Green PCR Master Mix (Applied Biosystems). Triplicate PCR reactions were carried out in 96-well plates using a 7300 Real-Time PCR System. The conditions were 50 °C for 2 min, 95 °C for 10 min, followed by 40 cycles of 95 °C for 15 s (s) and 60 °C for 1 min. The mRNA level of the indicated gene was normalized to that of the invariable control gene, 36B4. −∆ cycle threshold (C_T_) values were obtained by subtracting the C_T_ value of 36B4 (an invariable control gene) from the C_T_ values of the indicated genes, which showed the difference between the expression levels of 36B4 and the indicated gene.

### 2.8. Protein Extraction and Western Blot

To prepare the total hepatic lysate, approximately 100 mg of liver tissue was homogenized in 1 mL of liver lysis buffer (50 mM Hepes, 10 mM EDTA, 10% glycerol, 1% NP-40 and 1% Triton X-100, 100 mM NaF, 1 mM sodium molybdate, 1 mM sodium β-glycerophosphate, 5 mM sodium orthovanadate, 1.9 mg / ml aprotinin, 5 μg/mL leupeptin, 1 mM benzamide, 2.5 mM PMSF, pH 8.0), using a tissue homogenizer (Tissue Master 240, OMNI International, Kennesaw, Georgia) at 6000 × rpm. After the homogenization, this tissue lysate was put on ice for at least 20 min, and then centrifuged at 20,000× *g* for 15 min. The supernatant (total hepatic lysate) was transferred to a fresh microcentrifuge tube and stored at −80 °C until being used.

The protein concentration in the total hepatic lysate was determined using the BCA kit. Loading buffer (0.5 M Tris-HCl, 30% glycerol, 10% SDS, 0.012% bromophenol blue, 100 mM DTT, pH 6.8) and deionized water were added to the lysates to adjust the final protein concentration to 2 µg/uL before protein samples were heated at 100 °C for 5 min. The proteins in the total hepatic lysate (60 μg/lane) were separated on an 8% or 10% SDS-polyacrylamide gel, and transferred to a polyvinylidene difluoride (PVDF) membrane (GE Healthcare, Cat #10600021) in a transfer buffer (2 mM Tris-Base, 19.2 mM glycine, 5% methanol, pH 8.0). The membranes were blocked in TBST (50 mM Tris-base, 150 mM NaCl, 0.1% Tween 20, pH 7.5) with 5% dry milk at room temperature for 1 h. The blocked membranes were incubated in TBST buffer containing the indicated primary antibodies at 4 °C for overnight. The names, catalog numbers, their dilutions, and manufacturers of these antibodies are listed in Appendix A. Bound primary antibodies were detected using horseradish peroxidase-conjugated goat anti-rabbit IgG diluted 1: 2000 (# 7074, Cell Signaling Technology) after incubation at room temperature for 1 h. The membrane with bound secondary antibody was visualized using chemiluminescence, and exposed to X-ray films, which were developed using a Konica SRX 101A Film Processor. Image J was used to measure the signal intensity of the protein bands. The ratio of intensities of indicated protein-to-β-actin bands in the same sample was calculated and used for quantification.

### 2.9. Statistics Analysis

Statistical analyses were performed with GraphPad Prism 8.0 (GraphPad Software, Inc., San Diego, CA, USA). Student’s *t*-test with 95% confidence interval was used to compare to two groups. One-way ANOVA with least significance difference post hoc statistical analysis was used to compare the results of more than two groups. If necessary, a natural log transformation was performed before analysis. Data are presented as means ± S.E.M. The difference was considered significant when the *p*-value was less than 0.05.

## 3. Results

### 3.1. The Effects of Dietary VA and Fat Statues on BM, Food Intake, and PBG Levels in ZL and ZDF Rats

To investigate the effects of VA status on obesity and T2D development, ZL and ZDF rats at weaning (3 weeks of age) were fed a VAD, VAM or VAS diet with BF or HF content for 8 weeks. Their BM, food intake, and PBG levels were measured weekly. As shown in Figure 1A, ZL rats fed the VAS-BF diet had higher BM than those fed the VAD-BF or VAM-BF diet at weeks 6, 7 and 8, whereas ZL rats fed the VAM-BF diet had higher BM than those fed the VAD-BF at weeks 7 and 8. On the other hand, ZL rats fed the VAD-HF diet started to have a lower BM than those fed the VAS-HF diet at week 4 and those fed the VAM-HF diet at week 6, as shown in Figure 1B. ZDF rats fed the VAS-BF diet started to have a higher BM than those fed the VAD-BF or VAM-BF diet at week 6, whereas the BM of ZDF rats fed the VAD-BF diet was not different from those fed the VAM-BF diet, as shown in Figure 1C. ZDF rats fed the VAD-HF diet started to have lower BM than that fed the VAS-HF diet at week 4, and that fed the VAM-HF diet at week 6 as shown in Figure 1D. The ZDF rats fed the VAS-HF diet started to have higher BM than that fed the VAM-HF diet at week 7. The BM gain of ZL rats fed the VAD-BF or VAD-HF diet stopped at week 6, whereas those fed the VAM-BF and VAM-HF diets slowed down and those fed the VAS-BF and VAS-HF diets continued to gain BM throughout the experimental period. The BM of ZDF rats fed the VAS-BF or VAS-HF diet started to become significantly higher than those fed the VAM-BF and VAD-BF or VAM-HF and VAD-HF diets at week 6 or 7, respectively. These results demonstrate the role of VA in the BM gain of both ZL and ZDF rats.

The weekly food intake of ZL rats fed the VAS-BF diet was lower than that of the rats fed the VAD-BF diet and that of the rats fed the VAM-BF diet at weeks 2 and 7, respectively, as shown in Figure 2A. The weekly food intake of ZL rats fed the VAD-HF diet was lower than that fed the VAS-HF diet at weeks 7 and 8, as shown in Figure 2B. The weekly food intake of ZDF rats fed the VAD-BF diet was not different from that of the rats fed the VAM-BF or VAS-BF diet, as shown in Figure 2C. The weekly food intake of ZDF rats fed the VAD-HF diet was lower than that of the rats fed the VAM-HF diet and that of the rats fed the VAS-HF diet at weeks 6–8 and 7–8, respectively, as shown in Figure 2D. There appeared to be a trend of decrease in food intakes after the performances of the ITT (week 6.5) and OGTT (week 7.5) (Figure 2).

The PBG levels of ZL rats fed the VAD, VAM or VAS diet were not significantly different from each other in the BF (Figure 3A) or HF (Figure 3B) conditions, except for the ZL rats fed the VAS-BF diet at week 8, which was higher than those fed the VAD-BF or VAM-BF diet. The weekly BPG levels of ZDF rats fed the VAS-BF diet started to be higher than that of the rats fed the VAD-BF or VAM-BF diet at weeks 5 to 7, and that of the rats fed the VAM-BF diet at week 8, as shown in Figure 3C. Interestingly, as shown in Figure 3D, ZDF rats fed the VAM-HF or VAS-HF diet had higher PBG levels than those fed the VAD-HF diet at week 4, which is one week earlier than ZDF rats fed the BF diet.

### 3.2. The Effects of Dietary VA and Fat Statues on the Liver/BM and WAT/BM Parameters of ZL and ZDF Rats

The ratios of the liver to BM (liver/BM) and the epididymal WAT to BM (WAT/BM) were calculated to determine the effects of VA status on the body compositions of treated ZL and ZDF rats. As shown in Figure 4A, at the time of sacrifice, the liver/BM ratio of ZDF rats fed the VAD-BF or VAM-BF diet was significantly lower than that fed the VAS-BF diet. The liver/BM ratio of ZDF rats fed the VAD-HF diet was lower than that of the rats fed the VAS-HF diet. The liver/BM ratio of ZDF rats fed the VAS-BF was higher than that of the rats ZL rats fed the VAS-BF diet and the ZDF rats fed the VAS-BF diet.

As shown in Figure 4B, the WAT/BM ratios of ZDF rats were higher than those of the ZL rats fed the same diets. The HF diets increased the WAT/BM ratio in ZDF rats as the WAT/BM ratio of ZDF rats fed the VAS-HF diet was higher than that of the rats fed the VAS-BF diet. The WAT/BM ratio of ZDF rats fed the VAD-HF diet was lower than that of the rats fed the VAS-HF diet, which is similar to the pattern of the liver/BM ratio. The ratios of ZL and ZDF rats fed the VAM-BF diet were, respectively, not different from that of those rats fed the VAM-HF.

### 3.3. The Effects of Dietary VA and Fat Statues on the BPG Levels of ZL and ZDF Rats in ITT and OGTT

Animals were fasted overnight before the performances of ITT and OGTT, which were generally conducted at around 10–11 AM. As shown in Figure 5A, after insulin administration, the PBG levels of ZL rats fed the VAD-BF or VAM-BF, and VAS-BF diets dropped until 15 or 20 min, respectively. The PBG levels of ZL rats fed the VAD-HF and VAM-HF or VAS-HF diets declined until 30 and 15 min, respectively, as shown in Figure 5B. The PBG levels of ZDF rats fed the VAM-BF and VAS-BF diets dropped until 15 and 5 min, respectively, as shown in Figure 5C. The drop of PBG levels in ZDF rats fed the VAD-BF started at 10 min and stopped after 20 min. Figure 5D showed that the PBG levels in ZDF rats fed the VAD-HF and VAM-HF diets, or VAS-HF diets reached their nadirs at 15 or 20 min, respectively. The PBG level of ZDF rats fed the VAS-HF diet was significantly higher than that of the rats fed the VAD-HF and VAM-HF diets, indicating insulin resistance in them.

As shown in Figure 6A, the PBG level of ZL rats fed the VAS-BF diet peaked at 10 min, whereas the PBG of ZL rats fed the VAM-BF and VAD-BF diets peaked at around 30 min. The PBG levels of ZL rats fed the VAS-BF and VAS-HF diets were significantly higher than those of the rats fed VAD-BF or VAM-BF diet at 20 to 60 min (Figure 6A) and those of the rats fed the VAM-HF or VAD-HF diet at 10 and 60 min (Figure 6B), respectively. The PBG levels of ZL rats fed the VAD-HF diet was lower than those of the rats fed the VAM-BF diet at 20 to 60 min (Figure 6B).

The PBG levels of all ZDF rats peaked at 60 min (except those of the rats fed the VAS-BF diet at 30 min) before dropping (Figure 6C, D). ZL at 10 to 60 min and ZDF rats at the 20–120 min fed the VAS-BF diets had a higher BPG than the ZL and ZDF rats fed the VAD/VAM-BF diets, respectively. The PBG levels of ZDF rats fed the VAS-BF diet were higher than those of the rats fed the VAM-BF and VAD-BF diets at 20 to 120 min, and those of the rats fed the VAM-BF at 10 min only, as shown in Figure 6C. The ZDF rats fed the VAM-HF diet had a lower PBG level than that of the rats fed the VAD-HF and VAS-HF at 20 min, and that of the rats fed the VAS-HF only at 30 and 120 min, whereas the BPG level of ZDF rats fed the VAS-HF diet was higher than that of the rats fed the VAD-HF and VAM-HF diets at 60 min, as shown in Figure 6D.

Figure 6E shows the area under the curve (AUC) values, calculated using the OGTT data. The AUG of ZL rats fed the VAS-BF diet was higher than that of the rats fed the VAD-BF and VAM-BF diets. The AUC of VAS-HF ZL rats was higher than that of the rats fed the VAM-HF diet, which in turn was higher than that of the rats fed the VAD-HF diet. The AUC values of ZDF rats fed the VAS-BF or VAS-HF diets were, respectively, higher than those of the rats fed the VAD-BF and VAM-BF, or VAD-HF and VAM-HF diets, demonstrating the improvement in glucose tolerance when the VA status is lowered in ZDF rats. When the AUG values of ZL and ZDF rats fed the same diet were compared, the ZDF rats had significantly higher AUG values than ZL rats, indicating the impaired glucose tolerance in the ZDF rats.

### 3.4. The Effects of Dietary VA and Fat Statues on the Venous Plasma Levels of Glucose, TAG, Total Cholesterol, Insulin, Glucagon, and Leptin in ZL and ZDF Rats

The levels of venous blood glucose, TAG, insulin, cholesterol, leptin, and glucagon in the accumulated venous blood ZL and ZDF rats fed a VAD, VAM or VAS diet with BF or HF diet were measured. A shown in Figure 7A, the plasma glucose levels of ZDF rats fed the VAD-HF, VAM-HF, and VAS-HF diets were, respectively, higher than those of ZL rats fed the same diets. However, the venous blood glucose levels of ZDF rats fed the VAD-BF, VAM-BF and VAS-BF diets were not significantly different among these three dietary groups, which is different from the weekly PBG pattern shown in Figure 3C. The glucose levels of ZL rats fed the VAM-BF diet were significantly higher than those of the rats fed the VAD-BF diet. The venous blood glucose level of ZDF rats fed the VAD-HF was significantly lower than that of the rats fed the VAM-HF or VAS-HF diet.

The venous blood TAG level of ZL rats fed the VAS-BF diet was higher than that of the rats fed the VAM-BF or VAD-BF diet, as shown in Figure 7B. The TAG level of ZL rats fed the VAD-HF diet was lower than that of the rats fed the VAM diet, which was lower than that of the rats fed the VAS-HF diet. The TAG level of ZDF rats fed the VAD-BF diet was lower than that of the rats fed the VAS-BF diet, whereas those fed the VAM-BF was not different from these two. The TAG levels of ZDF rats fed the VAD-HF, VAM-HF and VAS-HF were not different, showing the impact of the HF diet. The TAG level of ZDF, but not ZL, rats was higher than that of ZL rats fed the same diet. Interestingly, the TAG levels of ZDF rats fed the VAS-BF diet was higher than that of rats fed the VAS-BF diet.

Figure 7C shows that the plasma insulin levels of ZL rats were significantly lower than that in ZDF rats fed the same diet. The venous blood insulin levels of ZL rats fed the BF and HF diets were not different among the three groups with different VA statuses and fat contents. The plasma insulin levels of ZDF rats fed the VAD/VAM-BF, or VAD/VAM-HF diets were significantly lower than those of ZDF rats fed VAS-BF or VAS-HF diets, respectively. These results indicate that the dietary VA may directly affects plasma insulin levels in ZDF rats, regardless the dietary fat contents.

Figure 7D shows that the cholesterol levels of ZDF rats fed the indicated diets (except the VAD-HF diet) were higher than ZL rats fed the same diet. The plasma cholesterol levels in ZL rats were not affected by the VA statuses and dietary fat contents. The plasma cholesterol levels of ZDF rats fed the VAS-BF and VAS-HF diets were higher than the ZDF rats fed the VAD-BF or VAM-BF and VAD-HF or VAM-HF diets, respectively.

Figure 7E shows that the leptin levels of ZL rats fed the BF diet with different VA statues were similar. The leptin levels of ZL rats fed the VAD-HF and VAM-HF diets were lower than those of the rats fed the VAS-HF diet. The ZDF rats fed the VAD-BF and VAD-HF diets had lower leptin levels than that fed the VAS-BF and VAS-HF diets, respectively. On the other hand, the leptin levels of ZDF rats fed the VAM-BF and VAM-HF diets were not different. The leptin levels of ZL rats fed the indicated diets were significantly lower than those of ZDF rats fed the same diets. There is a significant increase in plasma leptin levels in ZL rats fed the VAS-HF diet compared with those fed the VAS-BF diet. Figure 7F shows that the glucagon levels of ZDF rats fed the VAS-HF diet were higher than those of ZL rats fed the VAS-HF diet and lower than those of ZDF rats fed the VAS-BF diet.

The HOMA-IR values were calculated based on the plasma glucose and insulin levels, as shown in Figure 7G. The ZL rats fed the BF diets with different VA statuses had similar HOMA-IR values, and the ZL rats fed the VAS-HF diet have higher HOMA-IR values than ZL rats fed the VAD-BF diet. The HOMA-IR values of ZDF rats fed VAS-BF or VAS-HF diets were, respectively, higher than those of ZDF rats fed the VAD-BF and VAM-BF, or VAD-HF and VAM-HF diets, respectively. The HOMA values of ZDF rats fed the VAM-HF or VAS-HF diet were, respectively, higher than those of ZDF rats fed the VAM-BF or VAS-BF diet. The HOMA-IR values of ZDF rats fed the indicated diets were higher than those of ZL rats fed the same diets, respectively. The ZDF rats fed the VAS-BF diet had lower HOMA-IR values than the ZDF rats fed the VAS-HF, which is the highest in all groups.

### 3.5. The Effects of VA and Dietary Fat Statuses on the mRNA Levels of Hepatic Genes for Glucose and Lipid Metabolism in ZL and ZDF Rats

To determine the underlying mechanism by which the dietary VA and fat statuses regulate glucose and fat metabolism, we measured the hepatic mRNA levels of *Gck*, *Srebp-1c*, *Srebp2*, *Fas*, *Acl*, and *Pck1* in ZL and ZDF rats fed a VAD, VAM or VAS diet with BF or HF content. Figure 8A shows that in both BF and HF conditions, the hepatic mRNA expression levels of *Gck* in ZL rats fed the VAD and VAM diets were lower than those in rats fed the VAS diets. In ZL rats, the *Gck* mRNA levels in the VAS-HF group were increased significantly compared with those in the VAS-BF group. Additionally, the *Gck* mRNA levels in ZDF rats fed the VAD-BF diet were significantly higher than those in ZL rats fed the VAD-BF diet. These results demonstrate that VA is needed to sustain the hepatic expression of *Gck* in ZL rats, and especially in HF diet condition. VA status regulates the hepatic expression of *Gck* mRNA.

Figure 8B shows that the mRNA levels of *Srebp-1c* in ZL rats fed the VAD-BF diet were lower than those of the rats fed the VAM-BF, VAS-BF and VAD-HF diets. The *Srebp-1c* mRNA levels in VAM-HF ZDF group were higher than those of the rats in the VAM-BF ZDF group. Additionally, the ZDF rats fed the VAM-HF diet had higher *Srebp-1c* mRNA levels than those of the ZL rats fed the VAM-HF diet. On the other hand, the *Srebp2* mRNA levels of ZL rats fed the VAD-BF diet were higher than those of ZL rats fed the VAM-BF, VAS-BF and VAD-HF diets (Figure 8C). The *Srebp2* mRNA levels of ZDF rats fed the VAS-BF diet were lower than those of ZDF rats fed the VAD-BF diet. In contrast, ZDF rats in all dietary groups had significantly lower *Srebp2* mRNA expression levels than ZL rats fed the same diets (except the VAS-HF diet). This demonstrates that VA signaling induces *Srebp-1c* and suppresses *Srebp2* mRNA expression levels in the BF condition, and does not affect *Srebp-1c* and *Srebp2* mRNA levels in the HF conditions.

For gluconeogenesis, the *Pck1* mRNA levels in ZL and ZDF rats fed the VAD-HF diet were higher than those in ZL and ZDF rats fed the VAM-HF diet, respectively, as shown in Figure 8D. Interestingly, the ZL rats fed the indicated diets had higher *Pck1* mRNA levels than the ZDF rats fed the same diet, respectively. The *Pck1* mRNA levels in ZL rats fed the VAM-BF diet or ZDF rats fed the VAS-HF diet were higher than those in ZL rats fed the VAM-HF diet or ZDF rats fed the VAS-BF diet, respectively.

For the lipogenic genes, the *Fas* mRNA levels in ZDF rats fed the BF and HF diets (except the VAD-HF diet) were higher than those in ZL rats fed the same diet, respectively. The *Fas* mRNA levels in ZL rats fed the VAD-HF diet were higher than those fed the VAM-HF diet, whereas the levels in ZDF rats fed the VAS-HF diet were higher than those in rats fed the VAD-HF or VAM-HF diet, as shown in Figure 8E. The *Acl* mRNA levels in ZL rats fed the VAM-HF diet or ZDF rats fed the VAS-BF diet were, respectively, lower than those in ZL rats fed the VAM-BF diet or those in ZDF rats fed the VAS-HF diet, as shown in Figure 8F. The *Acl* mRNA levels of ZL rats fed the indicated diets (except VAD-BF and VAS-BF diets) were lower than those of ZDF rats fed the same diets.

### 3.6. The Effects of VA and Dietary Fat Statuses on the Protein Levels of Hepatic Genes for Glucose and Lipid Metabolism in ZL and ZDF Rats

To determine whether the changes of mRNA levels lead to alterations of the proteins, we analyzed the hepatic proteins levels FAS, ACL, IRβ, PEPCK, GCK, and β-actin (as the loading control). For the ZL rats, the hepatic ACL levels in ZL rats fed the VAD-BF diet were lower than those in ZL rats fed the VAS-BF diet, as shown in Figure 9A. The IRβ levels in ZL rats fed the VAD-BF and VAM-BF were slightly lower than those in rats fed the VAS-BF diet. The PEPCK levels in ZL rats fed the VAM-BF diet were lower than those in the rats fed the VAS-BF diet. The GCK mRNA levels of VAD-BF and VAM-BF were lower than those in rats fed the VAS-BF-diet. Figure 9B shows that the ACL levels in ZL rats fed VAD-HF diet were lower than those in rats fed the VAS-BF diet, which were lower than those in rats fed the VAM-HF diet. The IRβ expression level in ZL rats fed the VAS-HF diet was lower than that in rats fed the VAD-HF diet.

For ZDF rats, Figure 9C shows that the ACC and ACL levels in ZDF rats fed the VAD-BF and VAM-BF diets were significantly lower than those in rats fed the VAS-BF diet. The FAS and PEPCK levels in ZDF rats fed the VAD-BF diet were significantly lower than those in rats fed the VAS-BF diet. The IRβ levels in ZDF rats fed the VAD-BF and VAM-BF diets were slightly higher than those in rats fed the VAS-BF diet. Figure 9D shows that the ACC, FAS, and ACL levels in ZDF rats fed the VAD-HF and VAM-HF diets were significantly lower than those in rats fed the VAS-HF diet. The IRβ levels in the ZDF rats fed the VAM-HF diet were slightly higher than those in rats fed the VAS-HF diet. The GCK levels in ZDF rats fed the VAD-HF diet were significantly lower than those in rats fed VAM-HF and VAS-HF diets. These results show that VAS status suppresses IRβ levels, and supports ACC, FAS, ACL, and GCK in ZDF rats fed a HF diet.

## 4. Discussion

We found that the lowered dietary VA status can prevent the development of T2D and obesity in ZDF rats fed a BF or a HF diet. This prevention is associated with changes in the expression levels of hepatic genes involved in glucose and fatty acid metabolism.

ZDF rats develop T2D with high frequency in males [37]. The lack of leptin receptor signaling due to mutations leads to hyperphagia in ZDF rats [32,38,39]. This results in obesity and T2D. Our results shown in this paper reveal that hyperglycemia only occurs in the ZDF rats fed the VAS diet, but not in those fed VAD and VAM diets. The ZDF rats fed the VAD or VAM diet with a BF or HF condition have lower BM than those fed the VAS diet with the same fat content at the end of this 8 week study. We have reported previously that ZF rats fed a VAD diet for 8 weeks had lower BM and food intake than those fed a VAS diet [12]. A following pair-feeding study demonstrated that energy intake alone could not be enough to explain the reduction in BM in ZF rats fed the VAD diet [10]. The food intakes of ZDF rats fed the VAD, but not VAM, diet dropped at weeks 7 and 8 compared with those of rats fed the VAS diet in both BF and HF conditions. Therefore, the reduction in food intake alone probably is not sufficient to explain the correction of hyperglycemia development in ZDF rats fed the VAD and VAM diets. We have shown previously that ZL rats fed a VAD diet have lower respiratory exchange rates than those fed a VAS diet, demonstrating the higher catabolic state associated with a low VA status [13]. Whether changes in metabolic states and a reduction in food intake contribute to the correction of T2D and obesity in ZDF rats fed a VAD or VAM diet deserves further investigation.

We have reported that ZF rats fed a VAD diet started to have a lower BM than those fed VAS diet ad libitum at an earlier time point compared to the ZL rats fed a VAD diet [12], suggesting that ZF rats are more sensitive to the lack of dietary VA than ZL rats. This phenomenon appears to exist in ZDF rats as well. In this study, the ZL rats fed the VAD-BF have a lower BM than those fed the VAM-BF diet, which have a lower BM than those fed the VAS diet (Figure 1A) on weeks 7 and 8. However, the ZDF rats fed the VAD-BF or VAM-BF diet have similar BM values, which are lower than those fed the VAS-BF diet at the same time points (Figure 1C). ZL (Figure 1A), but not ZDF (Figure 1C), rats fed the VAM diets with BF are able to grow (at a lowered rate) at weeks 7 and 8. VAM-HF and VAS-HF ZL or ZDF rats have a significantly higher BM than VAD-HF ZL or ZDF rats after being on the diets for 6 weeks, respectively. These results demonstrate that ZDF rats are more sensitive to VA status in the BF condition. Interestingly, the HF diet appears to mask this effect. The ZDF rats fed the VAM-HF diet can continuously gain BM and have a higher BM than those fed the VAD-HF diet on weeks 7 and 8. These phenomena clearly show the anabolic roles of VA in supporting animal BM gain, and there is an interaction between the dietary fat content and VA status in this context. Whether the higher dietary fat content leads to more VA absorption or less VA usage in the body remains to be seen in the future. Nevertheless, the results shown in this paper demonstrate that we established a model to answer those questions.

The PBG level of ZDF rats fed the VAS-BF diet is higher than that of those fed the VAD-BF and VAM-BF diets at the end of the study (Figure 3). However, the venous blood glucose levels of ZDF rats fed the VAS-BF diet only show a higher trend, which did not reach statistical significance, compared with those fed the VAD-BF and VAM-BF diets (Figure 7). The discrepancy of the venous blood glucose and PGB levels in ZDF fed the BF diets is interesting. First, this may be caused by the synthetic diet that we used to formulate the VAD, VAM and VAS diets. The ideal diet to induce diabetes in ZDF rats is Purina #5008 [40], which was not used in our study. In addition, the ZDF rats might develop peripheral hyperglycemia before the central one, which leads to the presence of hyperglycemia in the tail tip whole blood, but not the venous blood. The hyperinsulinemia observed in Figure 7C may be sufficient to control the glucose levels in the venous blood, but not in the peripheral regions. As far as we know, this is the first time that this phenomenon is reported in ZDF rats. Further studies are needed to test this hypothesis and determine the underlying mechanisms.

It is interesting to note that ZL rats fed the HF diets do not have higher TAG levels than the ZL rats fed the BF diet with the same VA status (Figure 7B). This phenomenon is also partially present in ZDF rats except for the ZDF rats fed the VAS-BF diet, which have higher TAG levels than the ZDF rats fed the VAS-HF diet. The feeding of a HF diet does not increase the plasma TAG level further in ZL and ZDF in the current experimental setting. It suggests a possible system that may prevent the rise of venous TAG levels in Zucker rats in a HF setting. We have reported that VA deficiency leads to reductions in plasma TAG in ZL and ZF rats fed a diet with BF [12], which matched our observation in this paper in ZL and ZDF rats. In addition, VA deficiency has been shown to reduce the hepatic lipogenesis and cholesterol synthesis [41,42]. The elevation of VA is associated with hepatic cholesterol, fatty acid, and TAG contents in rats [43]. Our observation that the lowered VA status leads to reduced plasma cholesterol levels in ZDF, but not ZL rats, is very interesting. It appears that the high plasma cholesterol level in ZDF rats requires sufficient signal from VA to maintain. When the VA status drops, the plasma cholesterol level decreases. Since we only measured total cholesterol, additional experiments are needed to determine the underlying mechanisms that lead to this reduction.

The dietary VA and fat statuses affected the expressions of hepatic genes involved in glucose and fatty acid metabolism at both mRNA and protein levels. We have reported previously that the hepatic *Gck* mRNA levels in ZL rats fed a VAD diet with BF content are lower than that fed a VAD diet [12,19]. The *Gck* mRNA and protein levels in this set of ZL rats fed the VAD-BF diet are lower than those of rats fed the VAS-BF diet, matching the previous results well. However, the reduction in *Gck* mRNA levels in ZL rats fed the VAD-HF and VAM-HF diets dose not result in a significant reduction in the hepatic GCK protein levels, demonstrating the impacts of dietary fat content on the GCK protein expression. We have reported the reduction in *Srebp-1* expression in ZL and ZF rats fed a VAD diet with BF content [12], which can be seen in this paper in the ZL rats, but not ZDF rats. Interestingly, the ACL protein levels in ZL rats fed the VAD/VAM-BF and VAD/VAM-HF are lower than those of rats fed the VAS-BF and VAS-HF, respectively. We have also reported that the ACL protein levels in ZL fed a VAS diet are higher than those of rats fed a VAD diet [12]. However, the mRNA levels in these groups do not show the same patterns. On the other hand, the *Acl* mRNA and ACL protein levels in ZDF rats fed the HF diet with different VA status match very well. Whether these discrepancies of *Gck* and *Acl* mRNA and their protein levels are due to differences of translation or protein degradation remains to be determined.

It is interesting to note that some of the regulations of mRNA and protein levels by VA status can only be observed in ZL or ZDF rats. For example, the *Gck* mRNA regulations by VA are only observed in ZL, but not ZDF rats, fed the BF or HF diets. The impacts of VA status on the GCK proteins can be observed in ZL rats fed the BF, but not the HF diets, and in ZDF rats fed the HF, but not the BF diets. All these show the complex interactions among the VA status, Zucker rat genotype, and dietary fat contents. It is sufficient to say that there is an interaction between VA signaling and leptin signaling, which ZDF rats lost. In addition, VA signaling may also interact with the dietary fat content, which also affects the insulin sensitivity based on the HOMA-IR and OGTT data. Lowered VA status prevented the increase in insulin resistance in the presence of HF. It is worth to note that the lowered GCK protein expression is associated with a reduction in plasma glucose levels and improvement in insulin sensitivity in ZDF rats fed the HF diets. Whether this reduction in ZDF rats helps to control glucose or it is an irrelevant change remains to be studied.

The HOMA-IR values and OGTT results clearly demonstrate that ZL and ZDF rats fed a VAD or VAM diet are, respectively, more insulin sensitive than those fed a VAS diet in the BF and HF conditions. The AUC values of ZDF rats fed the VAS-BF or VAS-HF diets were, respectively, higher than those of rats fed the VAD-BF and VAM-BF, or VAD-HF and VAM-HF diets, demonstrating the improvement of glucose tolerance when the VA status is lowered in ZDF rats. When the AUG values of ZL and ZDF rats fed the same diet were compared, the ZDF rats had significantly higher AUG values than ZL rats, indicating the impaired glucose tolerance in the ZDF rats. The similar improvements in insulin sensitivity in VAM groups clearly demonstrate that we can study the effects of VA status on T2D development in ZDF rats without the development of VA deficiency in rats. One seminal discovery in the treatment of diabetes happened 100 years ago and is the discovery of insulin [44]. Before that, dietary manipulation had been use to control blood glucose in patients with diabetes [45]. VA was also discovered less than 110 years ago [9]. With more advances in our understanding of the functional mechanisms of insulin and VA [46], it is time for us to reveal the roles of VA in the development of T2D. The VAM diet and ZDF rats can become a useful model to study the role of VA in the development of insulin resistance, obesity and T2D.

It has been known for a while that hypervitaminosis A (excessive amount of VA intake) leads to the elevation in total lipids in the liver, intestine and brain, but not in the heart, whereas it reduces the total lipids in the kidney [47]. In human subjects with acne, the use of isotretinoin (13-cis RA) results in hypertriglyceridemia [48]. In those patients with acute promyelocytic leukemia, treatment with RA leads to weight gain, and increases in blood triacylglycerol and cholesterol levels [49,50]. All these show that the excessive activation of VA signaling system may have detrimental effects on the lipid metabolism in animals and human subjects. Whether the VA metabolism and its signaling system have been elevated in ZDF rats and whether this leads to the development of T2D in them are open questions begging for answers.

Our study also inherited certain limitations. First, we tried to conduct ITT and OGTT studies, while also trying to collect food intake data. In addition, the rats were housed in groups with 1 to 3 per cage. All these might have reduced the accuracy of the food intake data. We plan to conduct pair-feeding experiments in the future to obtain more conclusive results and more insights into the mechanisms by which VA signaling interacts with dietary fat and Zucker rat genotype. Then, animals probably should be housed individually to increase accuracy. Second, the custom-made and synthetic basal diets used in this study were designed to test the impact of dietary fat contents (BF and HF). They were not built on the Purina #5008 diet, which is the preferred diet to induce T2D development in ZDF rats [40]. This may have limited the potency of the diet to induce T2D phenotypes in ZDF rats. Future dietary studies will be conducted in a diet that is much closer to the Purina #5008 to minimize impacts of any other factors not considered.

## 5. Conclusions

In summary, we successfully prevented the development of T2D and obesity in ZDF rats through lowering dietary VA statuses. All these occur in both BF and HF conditions using custom-made and synthetic diets. The increase in the dietary fat content from BF (22% energy) to HF (60% energy) appears to reduce the need of dietary VA to support anabolism in ZL and ZDF rats in the current experimental settings. In addition, we established a VAM dietary model to study the impacts of VA on glucose and lipid metabolism, and diabetes development in ZDF rats without introducing a VAD diet. This will help us to elucidate the underlying mechanisms by which VA signaling interacts with dietary components and rat genotypes to control the glucose and lipid metabolism in normal and T2D states.

## Figures and Tables

**Figure 1 biomolecules-12-00528-f001:**
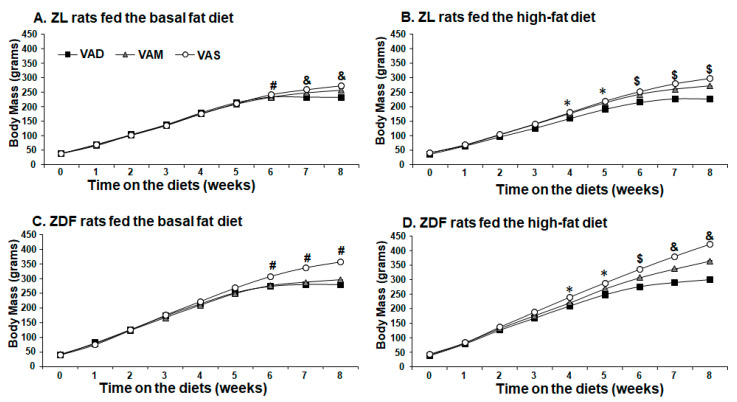
**Body Mass of ZL or ZDF rats fed a VAD, VAM or VAS diet with BF or HF content for 8 Weeks**. The body mass of ZL and ZDF rats was measured weekly. Data were expressed as mean ± SEM; *n* = 5–6 per dietary group; * For VAD < VAS; # for VAD/VAM < VAS; $ for VAD < VAM/VAS; & for VAD < VAM < VAS using one-way ANOVA at the time point indicated; all *p* < 0.05.

**Figure 2 biomolecules-12-00528-f002:**
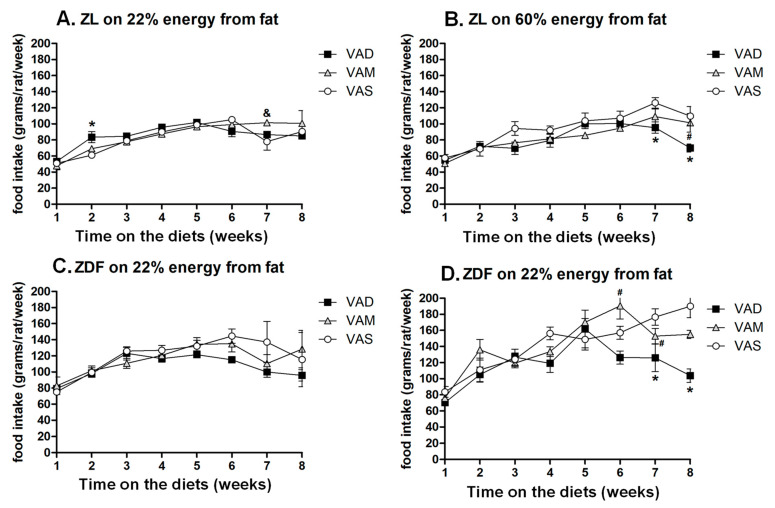
**Food intake of ZL or ZDF rats fed a VAD, VAM or VAS diet with BF or HF content for 8 Weeks**. The average amounts of food intake in the ZL or ZDF rats fed the indicated diets were measured weekly. Data are expressed as mean ± SEM; *n* = 5–6 per group; * VAD vs. VAS, # VAD < VAM, & VAM > VAS; all *p* < 0.05.

**Figure 3 biomolecules-12-00528-f003:**
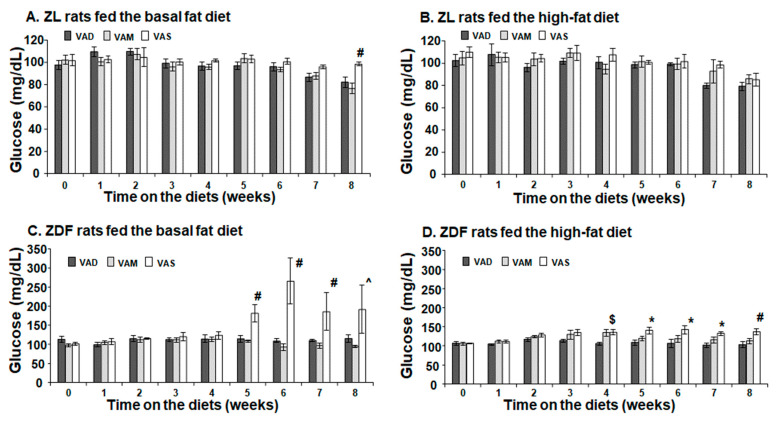
**Peripheral blood glucose levels of ZL or ZDF rats fed a VAD, VAM or VAS diet with BF or HF content for 8 Weeks.** The PBG levels from rat tail tips were measured weekly. Data are expressed as mean ± SEM; *n* = 5–6 per group; * For VAD < VAS; ^ for VAM < VAS: # for VAD/VAM < VAS; $ for VAD < VAM/VAS; all *p* < 0.05.

**Figure 4 biomolecules-12-00528-f004:**
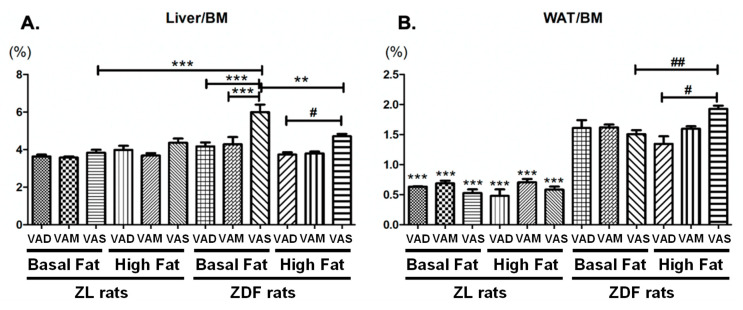
**The ratios of liver/BM (A) and WAT/BM (B) were calculated and compared between VAD/VAM/VAS-BF/HF ZL and ZDF rats.** The ratios of liver/BM and WAT/BM were calculated and compared. Data are represented means ± SEM, *n* = 5–6 per group. For the liver/BM, ** *p* < 0.01, *** *p* < 0.001 VAS-BF ZDF rats vs. other groups; # *p* < 0.05, ## *p* < 0.01 VAS-HF ZDF rats vs. other groups. For the WAT/BM, ** *p* < 0.01, *** *p* < 0.001 ZL rats vs. ZDF rats in same diet; # *p* < 0.05, ## *p* < 0.01 VAS-HF ZDF rats vs. other groups.

**Figure 5 biomolecules-12-00528-f005:**
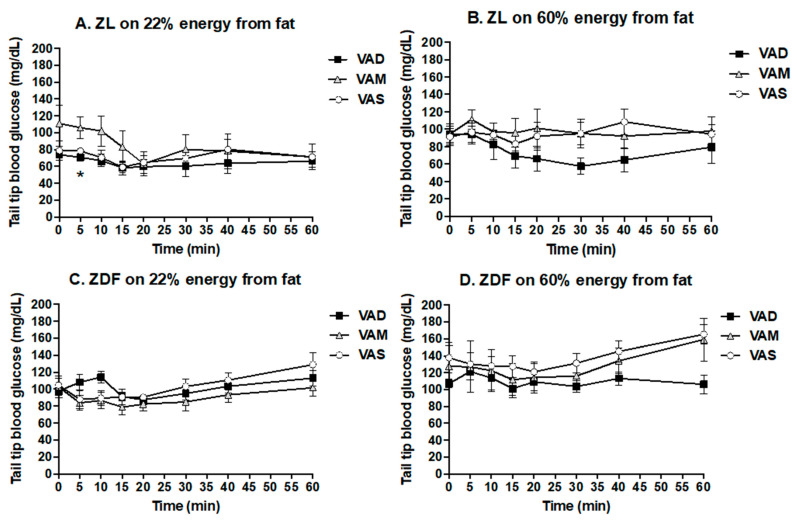
**The peripheral blood glucose levels of ZL and ZDF rats fed a VAD, VAM or VAS diet with BF or HF content for 6.5 weeks in the insulin tolerance test.** Data are expressed as mean ± SEM, *n* = 5–6 per group; * *p* < 0.05 for comparing the VAS-BF group with the VAD-BF and VAM-BF group.

**Figure 6 biomolecules-12-00528-f006:**
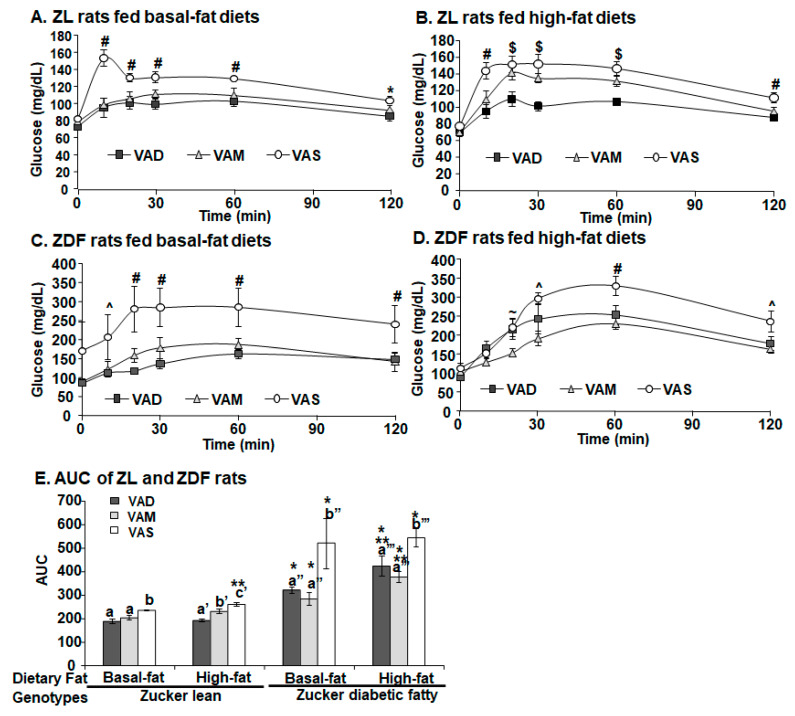
**The peripheral blood glucose levels of ZL (A,B) and ZDF (C,D) rats fed a VAD, VAM or VAS diet with BF or HF content in the oral glucose tolerance test, and AUG (E).** Data are presented as mean ± SEM; *n* = 5–6 per group. For (**A**–**D**), * for VAD < VAS; ^ for VAM < VAS; # for VAD/VAM < VAS; $ for VAD < VAM/VAS; ~ for VAM < VAD/VAS; all *p* < 0.05. For (**E**), a < b, a’ < b’, a” < b”, and a’’’ < b’’’ for comparing the groups in the same fat content and genotype using one-way ANOVA; * for comparing ZL and ZDF groups with the same diet; ** for comparing BF with HF groups within the same genotype using Student’s *t*-test.

**Figure 7 biomolecules-12-00528-f007:**
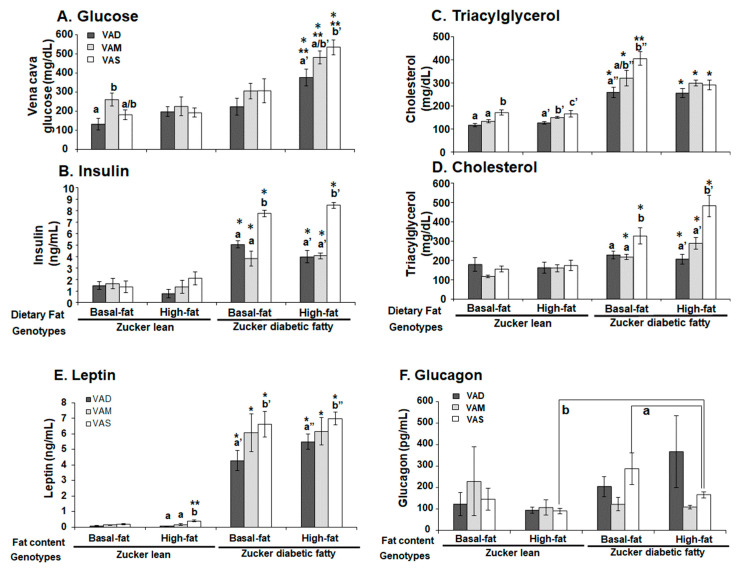
**The venous blood levels of glucose (A), triacylglycerol (B), insulin (C), cholesterol (D), leptin (E), and glucagon (F), and HOMA-IR (G) in ZL and ZDF rats fed a VAD, VAM or VAS diet with BF or HF content.** For (**A**–**E**,**G**), a < b, a’ < b’, a” < b”, and a’’’ < b’’’ for comparing the groups in the same fat content and genotype using one-way ANOVA; * for comparing ZL and ZDF groups with the same diet; ** for comparing BF with HF groups within the same genotype; all *p* < 0.05. For F, a for comparing the BF and HF conditions with the same rat genotype and VA status; b for comparing the ZL and ZDF rats in the same VA status and dietary fat content.

**Figure 8 biomolecules-12-00528-f008:**
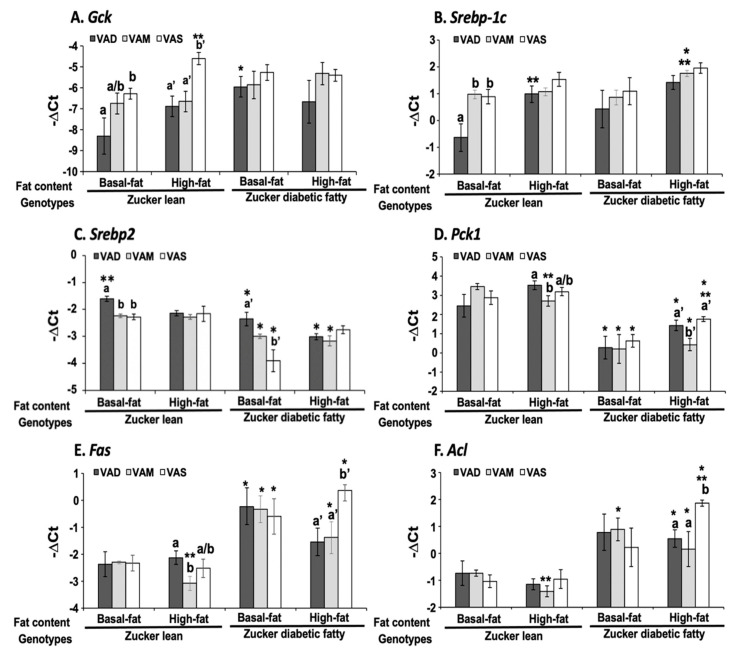
**The mRNA levels of Gck (A), Srebp-1c (B), Srebp2 (C), Pck1 (D), Fas (E), and Acl (F) in the liver of ZL and ZDF rats fed a VAD, VAM or VAS diet with BF or HF content**. Total RNA was extracted and subjected to real-time PCR analysis. The results are presented as means ± SEM of the -∆CT, and n = 5–6 per group. For A, a < b, and a’ < b’; for B, a < b; for C, a > b, a’ > b’; for D, a > b, and a’ > b’; for E, a > b and a’ < b’; for F, a < b, for comparing the ZL or ZDF rats fed the diets with different VA status with the same fat content using one-way ANOVA. All shown * for comparing ZL and ZDF groups fed the same diet; ** for comparing the ZL or ZDF rats fed the BF diet with those fed the HF diet with the same VA status; all p < 0.05.

**Figure 9 biomolecules-12-00528-f009:**
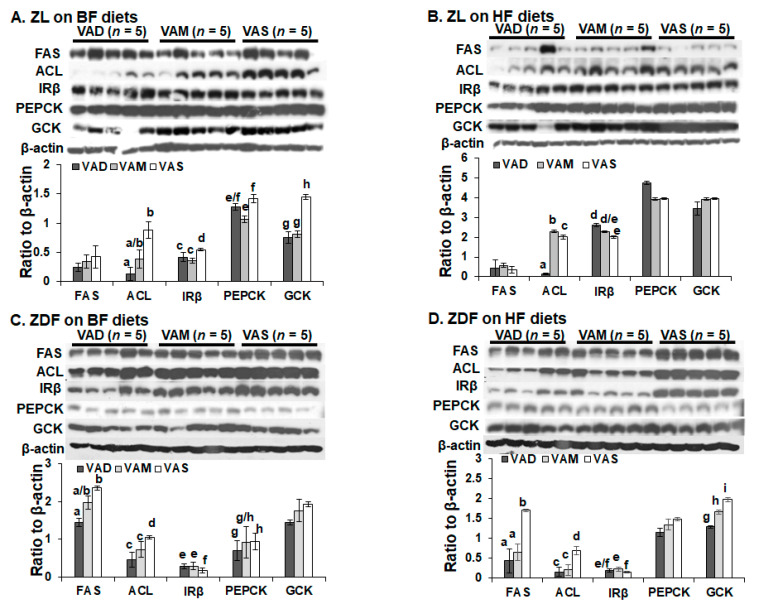
**The hepatic levels of FAS, ACL, IRβ, PEPCK, and GCK in ZL (A,B) and ZDF (C,D) rats fed a VAD, VAM or VAS diet with BF (A,C) or HF (B,D) content for 8 weeks.** Proteins (60 μg/well) in whole tissue lysates were separated and analyzed using Western blot analysis. The ImageJ Software was used to determine the densitometry of the indicated protein band. The results are presented as means ± SEM of the ratio of the indicated protein to β-actin (n = 5–6 per group). (A) a < b, c < d, e < f, and g < h for comparing the groups in the same fat content and genotype; (B) a < c < b, and e < d; (C) a < b, c < d, f < e, and g < h; (D) a < b, c < d, f < e, g < h < i for comparing the groups in the same fat content and genotype using one-way ANOVA; all p < 0.05.

**Table 1 biomolecules-12-00528-t001:** Protein, carbohydrate, fat, VA and caloric content of each diet used in the study.

Fat Content	Basal Fat			High Fat		
VA Statuses Catalog NO.	VAD #TD. 190039	VAM #TD. 190040	VAS #TD. 190041	VAD #TD. 190036	VAM #TD. 190037	VAS #TD. 190038
Protein (% kcal)	20.1	19.5	19.5	18.9	18.3	18.3
Carbohydrate (% kcal)	57.8	57.9	57.9	21.3	21.4	21.4
Fat (% kcal)	22.1	22.7	22.7	59.7	60.3	60.3
VA (mg/kg)	0	0.35	4.0	0	0.35	4.0
Calories (Kcal/g)	3.8	3.8	3.8	5.1	5.1	5.1

Note: VAD, VA deficient (0 mg retinyl palmitate/kg diet); VAM, VA marginal (0.35 mg retinyl palmitate/kg diet); VAS, VA sufficient (4.0 mg retinyl palmitate/kg diet).

## Data Availability

Not applicable.

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
