# Peer review of "Reduction in the Dietary VA Status Prevents Type 2 Diabetes and Obesity in Zucker Diabetic Fatty Rats"

_biomolecules, 2022, doi:10.3390/biom12040528_

Round 1

Reviewer 1 Report

     The manuscript titled “Reduction of the dietary VA status prevents type 2 diabetes and 2

obesity in Zucker diabetic fatty rats” is a good study. The authors concluded that the reduced   dietary Vitamin A  status in both Basal Fat  and High fat conditions prevents type 2 diabetes and  obesity in Zucker diabetic fatty  rats. The parameters including  hepatic mRNA and proteins levels using real- 17 time PCR and Western blot methord was good .The results are very interesting  promoting further scope .The presentation of the results were very good. The authors has discussed the results with the proper references. They have also established a VAM dietary model to study the impacts of VA on glucose and lipid metabolism, and diabetes development in ZDF rats without introducing a VAD diet.  Which will help  to elucidate underlying mechanisms by which VA signaling interacts with dietary components and rat genotypes to control the glucose and lipid metabolism .Overall it was a good manuscript.

Strength of the study

The experimentation procedure

The Robust parameters used

The interpretations of the results

Clinically relevant

Weakness- NIL

major recommendations for the improvement of the manuscript:

The study could be improved by adding histological analysis of the pancreas or liver since the study has been related to metabolism

minor recommendations for the improvement of the manuscript:

Only the grammatical errors could be rectified

Author Response

Responses to the Reviewer #1’s Comments

Manuscript NO.: Biomolecules-1620239

Manuscript title: " Reduction of the dietary VA status prevents type 2 diabetes and obesity in Zucker diabetic fatty rats"

The authors would like to thank the reviewer #1 for the time and efforts, especially those excellent comments. We have revised the manuscript according to the reviewer’s comments. Here are our responses to the comments line-by-line. The comments are shown first, which are followed by our responses in italics.

Please note that the words and sentences that have been revised are highlighted in yellow.

Reviewer #1

1.1 Major recommendations for the improvement of the manuscript: The study could be improved by adding histological analysis of the pancreas or liver since the study has been related to metabolism.

Responses: Thank you for the comments. The goal of study was to investigate the role of vitamin A (VA) in the development of obesity and type 2 diabetes in Zucker diabetic fatty (ZDF) rats. The study has been completed. Unfortunately, due to the limited funding, we did not order histological analysis of the liver and pancreas. In addition, these investigations can be done when we study the pathological changes in those organs. However, these were not our original goals. They are very useful data to obtain when we want to determine the VA’s impacts in those organs in the future. Moreover, we only were only given 10 days to revise the manuscript, which is not enough for us to find financial support and complete the analysis. We will remember to compete this task in the future studies, which we are actively looking for support at this time.

1.2 Minor recommendations for the improvement of the manuscript: Only the grammatical errors could be rectified.

Responses: Thank you for the suggestions. We have done our best to revise this manuscript and to avoid grammatical errors. Please see the revised manuscript for the changes.

Reviewer 2 Report

Chen et al. prepared a paper on Reduction of the dietary VA status prevents type 2 diabetes and 2 obesity in Zucker diabetic fatty rats....
The paper requires many editorial revisions.
The interlining is sometimes wider, sometimes wider, the text is sometimes not justified, the titles of subchapters at the bottom of the page or flowers like line 523 which is the beginning of the discussion hidden in the middle of the sentence...
The introduction is not very extensive, there is not much foreign data and lines 43-71 are descriptions of previous studies by the authors. The same applies to the discussion section... 14 citations for such an uncomprehensive bibliography is an awful lot!

There is a lack of decisiveness in the case of figures - one description as 1 and another as 1, but it concerns the same thing, i.e. weeks - it should be unified
fig 4 is not very readable in terms of the x-axis...maybe it should be divided into vam vas vad?
the results section starts very strangely with a graph? it does not lead the reader by the hand, what happened in turn and the reasoning behind the research - I do not see it here
How was the quality of RNA checked after isolation? No information, only that 2 micrograms were taken for reverse transcription reaction
one of the most important why biomolecules? please authors justify the journal, in my opinion it is not very fitting 

finally the attached original blots are not very nice and secondly they are exactly the same as in work, cut so you can't paste the same and claim it's different

Author Response

Responses to the Reviewer #2’s Comments

Manuscript NO.: Biomolecules-1620239

Manuscript title: " Reduction of the dietary VA status prevents type 2 diabetes and obesity in Zucker diabetic fatty rats"

The authors would like to thank the reviewer #2 for the time and efforts, especially those excellent comments. We have revised the manuscript according to the reviewer’s comments. Here are our responses to the comments line-by-line. The comments are shown first, which are followed by our responses in italics.

Please note that the words and sentences that have been revised are highlighted in yellow. Because additional citations have been added, the reference list should be considered as revised.

Reviewer #2

Chen et al. prepared a paper on Reduction of the dietary VA status prevents type 2 diabetes and 2 obesity in Zucker diabetic fatty rats..... The paper requires many editorial revisions.

2.1 The interlining is sometimes wider, sometimes wider, the text is sometimes not justified, the titles of subchapters at the bottom of the page or flowers like line 523 which is the beginning of the discussion hidden in the middle of the sentence...

Responses: Thank you for your comments. We could not find any “..flowers like line 523..”. The line 523 is the legend of Figure 9. This might be caused by the differences of computers of the authors and the reviewer. Nevertheless, we adjusted the interlining space to the single line format. Please see the revised manuscript for the changes.

2.2 The introduction is not very extensive, there is not much foreign data and lines 43-71 are descriptions of previous studies by the authors. The same applies to the discussion section... 14 citations for such an uncomprehensive bibliography is an awful lot!

Responses: Thank you for the comments. Unfortunately, there are not many studies to investigate the association of VA and the development of diabetes. We have shown the previous studies by us to introduce proofs to support our hypothesis, which was tested in this study. A systemic study to address the impacts of VA status on the development of type 2 diabetes in ZDF rats has not been done as far as we know. Nevertheless, we have added a paragraph (the fourth one) in the Introduction section to discuss the variation of different human studies and the effects of retinoic acid treatment on blood glucose in ZDF rats and ob/ob mice. Please see the revised manuscript for the changes.

2.3 There is a lack of decisiveness in the case of figures - one description as 1 and another as 1, but it concerns the same thing, i.e. weeks - it should be unified

Responses: Thank you for the comments. We have deleted “w” and unified the format of Figures 1, 2 and 3. Please see the revised manuscript for the changes.

2.4 fig 4 is not very readable in terms of the x-axis...maybe it should be divided into vam vas vad?

Responses: Thank you for the comments. We have revised the label of the x-axis. Hopefully, it becomes clear to our audience. Please see the revised manuscript for the changes.

2.5 the results section starts very strangely with a graph? it does not lead the reader by the hand, what happened in turn and the reasoning behind the research - I do not see it here

Responses: Thank you for pointing this out. We have added a sentence in beginning of the first paragraph of the Result section and revised the second sentence. Please see the revised manuscript for the change.

2.6 How was the quality of RNA checked after isolation? No information, only that 2 micrograms were taken for reverse transcription reaction

Response: Thank you for pointing this out. The quality of the RNA was determined using the ratio of absorbance at 260 nm and 280 nm. Only samples with the ratio higher than 1.8 were used. We have added this statement in the Materials and Methods. Please see the revised manuscript for the changes.

2.7 one of the most important why biomolecules? please authors justify the journal, in my opinion it is not very fitting

Responses: That is very interesting comment. The corresponding author was invited by the Biomolecules Editorial Office to contribute an article for the Special Issue "100th Anniversary of Insulin: Insulin Receptor Signaling in Health and Disease" in 2021. Insulin was discovered in 1921, and VA was also discovered in 1910s. It is a good idea to contribute an article to this journal. Since the role of VA in the development of type 2 diabetes fits the general framework of the special issue and the journal, I have decided to submit our research work to this journal.

2.8 finally the attached original blots are not very nice and secondly they are exactly the same as in work, cut so you can't paste the same and claim it's different.

Response: Our blots presented here were the original data from the films. We did not see any duplications. The quantification was done after the intensities of the indicated protein bands were normalized to that of the beta actin and compared as described in the Material and Method section. We do not know why the reviewer commented that “…they are exactly the same as in work, cut so you can't paste the same and claim it's different”. Please point out which protein bands are exactly same so that we can go back to check. We have supplemented the PowerPoint file of Figure 9, which contains the original blots. Please check.

Reviewer 3 Report

Introduction:

General comments:

The work has interesting results for understanding the metabolism of fat-soluble vitamins. However, some issues need to be clarified:

Introduction:

Lines – 73 – 80 The authors need make an objective of work. The text these paragraph showed results of work.

Material and methods

Line-148- 149 – the table 1 showed before of citation in the text. Please, adjust in the manuscript.

Lines 155 – 158 – The authors make a centesimal analysis of diets? Include methods in the manuscript.

Lines 155-158 - Was vitamin A analyzed in the basal diet (no added vitamin A)? What ingredients are used in the formulation of the diet? What is the source of lipids used in the diet to increase energy? Was it animal or plant lipid? Does this influence the amount of vitamin A in the diet? it is possible that the ingredients used in the formulation of the diet have vitamin A in their composition. Can the vitamin A present in the ingredients influence the results of the work?

Line 279 – adjust the “error”

Discussion

Line 523 – adjust the title “discussion”

Line  524 _ exclude “Here, we have fed ZL and ZDF rats a diet with different VA statuses (VAD, VAM, or VAS) and fat contents (BF or HF)”

Line 653 – The authors do not use a diet with pure ingredients. This can affect metabolism, as many ingredients added to diets may contain vitamin A. For example, vitamin A is present in oils and grains added to the diet. Can this vitamin A present in the ingredients affect the metabolism of rats?

Can too much vitamin A in diets cause toxicity? These can explain part of results. Please explain in the discussion.

Lines 644 – “In addition, the rats were housed in groups with 1 to 3 per cage. All these might have reduced the accuracy of the food intake data. We plan to conduct pair-feeding experiments in the future to get more conclusive results and more hinds in the mechanisms by which VA signaling interacts with dietary fat and Zucker rat genotype.” - What do the authors claim here is that the stocking rate is inadequate? Did this cause stress to the animals? Please, explain in the manuscript.

Conclusion

“The increase in the dietary fat content from BF (22% energy) to HF (60% energy) appears to reduce the need of dietary VA to support anabolism in ZL and ZDF rats.” The vitamin A is soluble in fat. For high energy of diets you need high fat addition in diet.  The reduction of need vitamin A can be due to vitamin A naturally present in the diet? please, explain in the discussion. Did the authors perform dietary vitamin A analyses? If yes, please add the results.

Author Response

Responses to the Reviewer #3’s Comments

Manuscript NO.: Biomolecules-1620239

Manuscript title: " Reduction of the dietary VA status prevents type 2 diabetes and obesity in Zucker diabetic fatty rats"

The authors would like to thank the reviewer #3 for the time and efforts, especially those excellent comments. We have revised the manuscript according to the reviewer’s comments. Here are our responses to the comments line-by-line. The comments are shown first, which are followed by our responses in italics.

Please note that the words and sentences that have been revised are highlighted in yellow. Because additional citations have been added, the reference list should be considered as revised.

Reviewer #3

General comments: The work has interesting results for understanding the metabolism of fat-soluble vitamins. However, some issues need to be clarified:

3.1 Introduction: Lines – 73 – 80 The authors need make an objective of work. The text these paragraph showed results of work.

Responses: Thank you for the comment. We have revised the paragraph to show the results of the work. Please see the revised manuscript for the changes.

3.2 Material and methods: Line-148- 149 – the table 1 showed before of citation in the text. Please, adjust in the manuscript.

Responses: Thank you for pointing out this. We have adjusted the location of table 1. Please see the revised manuscript for the changes.

3.3 Lines 155 – 158 – The authors make a centesimal analysis of diets? Include methods in the manuscript. Lines 155-158 - Was vitamin A analyzed in the basal diet (no added vitamin A)? What ingredients are used in the formulation of the diet? What is the source of lipids used in the diet to increase energy? Was it animal or plant lipid? Does this influence the amount of vitamin A in the diet? it is possible that the ingredients used in the formulation of the diet have vitamin A in their composition. Can the vitamin A present in the ingredients influence the results of the work?

Responses: Thank you for the comments. The diets were formulated by professionals in Envigo company. We have included the six types of diets as a supplementary information with the revision. No ingredient contained VA. There was not VA added in the basal diet. Lard was using as a lipid source, which does not provide any VA. Therefore, it is not possible for the VA presence in other ingredients to influence the results.

3.4 Line 279 – adjust the “error”

Response: Thank you for your comment. We have revised Figure 4 and adjusted its location to show the line number. Please see the revised manuscript for the changes.

3.5 Discussion Line 523 – adjust the title “discussion”

Response: Thank you for the comment. We have adjusted it. Please see the revised manuscript for the changes.

3.6 Line  524 _ exclude “Here, we have fed ZL and ZDF rats a diet with different VA statuses (VAD, VAM, or VAS) and fat contents (BF or HF)”

Responses: Thank you for your suggestion. The sentence has been deleted. Please see the revised manuscript for the changes.

3.7 Line 653 – The authors do not use a diet with pure ingredients. This can affect metabolism, as many ingredients added to diets may contain vitamin A. For example, vitamin A is present in oils and grains added to the diet. Can this vitamin A present in the ingredients affect the metabolism of rats?

Responses: Thank you for the comments. There are custom-made and synthetic diets. The basal diets do not contain any VA. We have added “custom-made and synthetic” in front of the “basal diets” to avoid any confusion. Please see the revised manuscript for the changes.

3.8 Can too much vitamin A in diets cause toxicity? These can explain part of results. Please explain in the discussion.

Response: Thank you for the comment. Yes, too much VA can cause detrimental effects. We have added one paragraph in the discussion section to address this. Please see the revised manuscript for the changes.

3.9 Lines 644 – “In addition, the rats were housed in groups with 1 to 3 per cage. All these might have reduced the accuracy of the food intake data. We plan to conduct pair-feeding experiments in the future to get more conclusive results and more hinds in the mechanisms by which VA signaling interacts with dietary fat and Zucker rat genotype.” - What do the authors claim here is that the stocking rate is inadequate? Did this cause stress to the animals? Please, explain in the manuscript.

Responses: Thank you for the comments. When the corresponding wrote the animal protocol of this study, animals were encouraged to be housed in groups rather than individually to increase some social activity and reduce their stresses. It appears that any experiment involved in food intakes, animals should be housed individually. We have added one sentence to discuss the benefits of housing animals individually. Please see the revised manuscript for the changes.

3.11 Conclusion “The increase in the dietary fat content from BF (22% energy) to HF (60% energy) appears to reduce the need of dietary VA to support anabolism in ZL and ZDF rats.” The vitamin A is soluble in fat. For high energy of diets you need high fat addition in diet.  The reduction of need vitamin A can be due to vitamin A naturally present in the diet? please, explain in the discussion. Did the authors perform dietary vitamin A analyses? If yes, please add the results.

Responses: Thank you for the comment. As we have explained previously, these are custom-made and synthetic diets, there was no VA from other ingredients other than those added to the diets. These diets were made and analyzed by professionals in Envigo. We have included the dietary fact sheets as supplemental materials. In addition, we have emphasized that these are custom-made and synthetic diets to avoid any confusion in the conclusion section. Please see the revised manuscript for the changes.

Round 2

Reviewer 2 Report

The authors have clarified the doubts